# A GIS Partial Discharge Defect Identification Method Based on YOLOv5

Yao Lu [ID], Zhibin Qiu [ID], Caibo Liao *, Zhibiao Zhou, Tonghongfei Li and Zijian Wu

Department of Energy and Electrical Engineering, School of Information Engineering, Nanchang University, Nanchang 330031, China
* Correspondence: lcb1752@126.com

**Abstract:** The correct identification of partial discharge types is of great significance to the stable operation of GIS. In order to improve the recognition accuracy and result of partial discharge, and to meet the requirements of real-time monitoring of GIS equipment, this paper proposes a GIS partial discharge defect recognition model based on YOLOv5. First, the GIS partial discharge simulation experiment is established to create the dataset of partial discharge PRPD map. Then, a YOLOv5-based GIS partial discharge defect recognition model is constructed, and different training methods are used to optimize the parameters of the model. By comparing with target detection models based on other deep learning methods, such as Faster-RCNN and YOLOv4, the YOLOv5 model discussed in the paper has significantly improved the recognition efficiency and recognition accuracy, in which mAP value is 95.89% and FPS is 28.89. In addition, the model can realize the distinction and identification of multiple PD types in a single PRPD map. At last, the YOLOv5-based GIS partial discharge defect identification model is applied to the test in a 500 kV substation. The model accurately determines the type of GIS partial discharge, which verifies the accuracy and validity of the model.

**Keywords:** gas insulated switchgear; YOLOv5; partial discharge; ultra-high frequency; defect detection; pattern recognition



## 1. Introduction

Gas insulated switchgear (GIS) has the advantages of small footprint, high level of integration, stable performance, and low maintenance. In recent years, GIS equipment has been widely used in power systems. Poor manufacturing process of GIS equipment, improper installation, vibration during transportation, etc., will lead to GIS components loose, resulting in GIS equipment failure [1]. Considering the economic and time cost of equipment maintenance, the detection of GIS insulation defects is of great significance. Partial discharge (PD) detection is one of the effective methods to analyze the operation and insulation state of GIS, and the damage caused by different types of PD to the GIS insulation is significantly different. Therefore, the effective identification of partial discharge types is of great importance to GIS equipment [2,3].

The PD signal identification of GIS mainly includes the time-based TRPD (time resolved partial discharge) mode and the PRPD (phase resolved partial discharge) mode based on phase analysis. PRPD pattern recognition technology has the advantages of more stable performance and smaller data scale, and is more widely used in practical applications [4]. Earlier, feature extraction methods based on PRPD pattern recognition mostly used statistical feature method, fractal feature method, image moment feature method, etc. [5]. However, the redundancy of feature parameters obtained by the above methods will not only affect the classification result but also increase the classification time. In recent years, with the successful application of deep learning in image recognition, speech recognition, semantic analysis, and other fields, Lenet5, Alexnet, deep residual networks, LSTM and other deep learning-based intelligent diagnosis methods for defective

faults have surpassed traditional machine learning methods and have gradually become a hot topic of research nowadays [6,7]. The literature [8] adopted MobileNets lightweight convolutional neural network for the classification of GIS partial discharge defect types. Song et al. published a complex partial discharge data source based on laboratory simulations and field data to verify the accuracy of deep CNN networks for GIS partial discharge pattern recognition under large data samples, and also demonstrate that the accuracy of GIS PD pattern recognition based on deep learning is much higher than that based on the traditional machine learning [9]. Han et al. proposed a novel adversarial learning framework, which resulted in a more robust feature representation, improved generalization of model training, and avoided overfitting when labeling small samples [10]. The literature [11] used generative adversarial networks (GAN) for data enhancement of partial discharge data and identified GIS partial discharge defects by building a MixNet deep learning model.

Deep learning has been widely used in fault diagnosis of electrical equipment. In the past, the identification of PD types was mostly based on classification methods, which required artificial extraction of PD features, such as discharge amplitude and time interval. In addition, the traditional GIS partial discharge defect identification based on classification methods may no longer be applicable in some special occasions. If multiple types of partial discharge defects occur in electrical equipment, the deep learning classification method used in the traditional partial discharge defect identification cannot distinguish multiple defects of the same type in one mapping, and it is difficult to distinguish multiple types of partial discharge defects. At the same time, traditional deep learning-based classification algorithms require complex parameter adjustment and data processing processes, while the application of deep learning-based object detection algorithms can achieve point-to-point recognition process, which greatly reduces detection time and saves detection costs.

The paper proposes a GIS partial discharge defect detection method based on YOLOv5 algorithm. First, a test circuit was established to simulate four typical partial discharge defects of GIS and a PRPD map dataset was built. Second, the paper constructs a GIS PD recognition model based on YOLOv5, including image preprocessing, image recognition and model parameter optimization, etc. Then, the YOLOv5 model was compared with the performance indicators of SSD, Faster-RCNN, YOLOv3, YOLOv4, and other models. Finally, the method was applied to the GIS online monitoring in a 500-kV substation, and the practicability and reliability of the method was discussed.

## 2. Acquisition of Partial Discharge Datasets

### 2.1. Partial Discharge Defect Model

The internal defects of GIS mainly include free metal particles inside the GIS, conductor tips, floating electrodes formed by poor conductor contact, surface contamination of solid insulators and internal air gaps. If these defects are not eliminated in advance, they will cause GIS metal particle discharge, corona discharge, floating electrode discharge, and insulation discharge respectively in the process of GIS operation, which will pose a great threat to the safe operation of GIS equipment. Therefore, based on the four common defects that may occur in manufacturing transportation and use of GIS in the process, a GIS experimental model was established to obtain partial discharge defects as shown in Figure 1.

During the partial discharge experiment, the influence of the external environment on the experimental equipment should be eliminated by increasing the sensor threshold. Before the start of the experiment, the ultra-high frequency (UHF) signal of GIS equipment under normal operation was measured, and the UHF sensor threshold range was adjusted to minimize the UHF acquisition value under normal operation of GIS, which can avoid to a certain extent the influence of environmental noise on the acquisition of GIS partial discharge signal, and then clear and accurate PRPD maps were obtained. This paper simulates four types of typical GIS partial discharge types. We placed four different discharge types of defects in the simulated gas chamber. To simulate corona discharge defect type, we used a copper pin pasted perpendicular to the high-voltage conductor on the GIS bus, the length of the copper pin is 2 mm, as shown in Figure 1a. To simulate

the floating electrode discharge defect type, we use insulating tape to suspend the epoxy between the high-voltage electrode and the metal housing, as shown in Figure 1b. The types of insulation discharge defects are simulated by setting bubbles of different sizes inside the insulator, as shown in Figure 1c. The metal particle discharge defect type is simulated by placing metal particles not exceeding 1 mm on the inner wall of the GIS enclosure, as shown in Figure 1d.

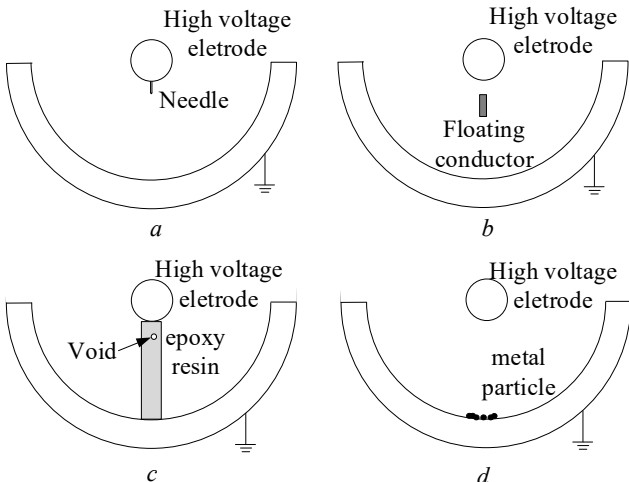

**Figure 1.** PD defect model of GIS. (**a**) metal tip defects, (**b**) floating electrode defects, (**c**) insulating void defects, (**d**) free metal particle defects.

### 2.2. Partial Discharge Detection Platform and Pressurization Method

This experiment is conducted in the partial discharge shielded room, and the partial discharge experimental platform is shown in Figure 2.

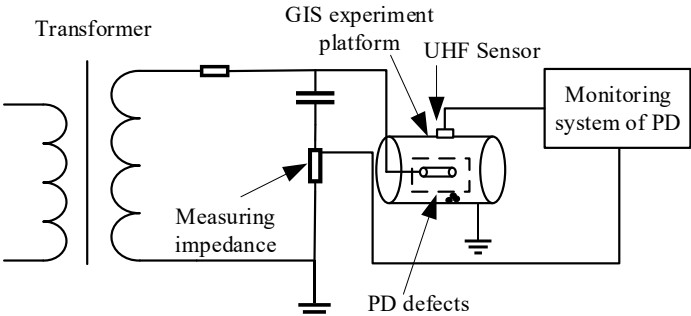

**Figure 2.** PD detection platform of GIS.

The platform mainly includes three parts: power supply system, test chamber, and measurement system. Among them, the power supply system consists of a test transformer and capacitor divider, which mainly provides the test cavity with industrial frequency AC voltage. The test chamber is a 550 kV true GIS section. The defect model is set in the test chamber, and the partial discharge signal generated by the defect model is transmitted to the partial discharge analyzer by the UHF sensor connected to the chamber for signal acquisition.

Set the test pressurization procedure as follows: raise the system voltage value to 318 kV and hold for 5 min; continue to raise the voltage to 550 kV and hold for 3 min; then raise the voltage to 740 kV and hold for 1 min; lower the voltage to 381 kV for partial discharge measurement, and lower the voltage to 0 after the partial discharge measurement.

Partial discharge defect data of GIS is collected by simulating partial discharge defects through experiments. Corona discharge is collected by gluing different lengths of copper pins perpendicular to the high-voltage conductor on the GIS bus. Copper pins of 1 mm, 2 mm, and 3 mm were set up to obtain PRPD patterns of corona discharge respectively. A total of 500 profiles were collected for each type of defect. Finally, 1217 valid PRPD profiles were

selected by manual screening. In simulating the type of floating electrode discharge defect, we use insulating tape to suspend different sizes of epoxy between the high voltage electrode and the metal housing. The width of the epoxy resin was set to 2 mm, 3 mm, and 4 mm, respectively. About 400 maps were collected for each type of defect by experiments. Finally, 1023 valid PRPD maps were obtained by manual screening. In the insulation discharge defect type simulation, insulators with simulated bubble diameters of 1 mm and 2 mm are installed on the experimental platform. A total of 500 partial discharge PRPD maps were collected for each type of defect through experiments, and finally, 948 valid PRPD maps were obtained through manual screening. In the simulation of free particle discharge defect types, 2 and 5 g of metal particles were placed in our experimental platform, and 600 maps were collected for each type of defect. Finally, 1053 valid PRPD patterns were obtained by manual screening.

### 2.3. Partial Discharge MAP

The PRPD model is the dominant feature representation, also known as the $\varphi$-$q$-$n$ model. It can visually represent the relationship between the working frequency phase $\varphi$ corresponding to the partial discharge pulse, the discharge volume $q$, and the number of discharges $n$ through images. In recent years, a large number of feature extraction methods have been used for partial discharge pulse phase distribution maps. Partial discharge maps were collected by simulating the GIS partial discharge environment. A total of 4241 effective GIS partial discharge maps are collected, including 1217 corona discharge maps, 948 insulation discharge maps, 1023 suspension discharge maps, and 1053 free particle discharge maps.

Since the horizontal and vertical coordinates of the PRPD map are fixed parameters, they have no effect on the shape of the map, and the background color of the PRPD map has little effect on the type of GIS partial discharge. Therefore, to facilitate target detection and classification of the map, the map images were grayed out. The result of the map after graying out is shown in Figure 3. The maps in the second row of the figure are the result of the first row of plots after graying.

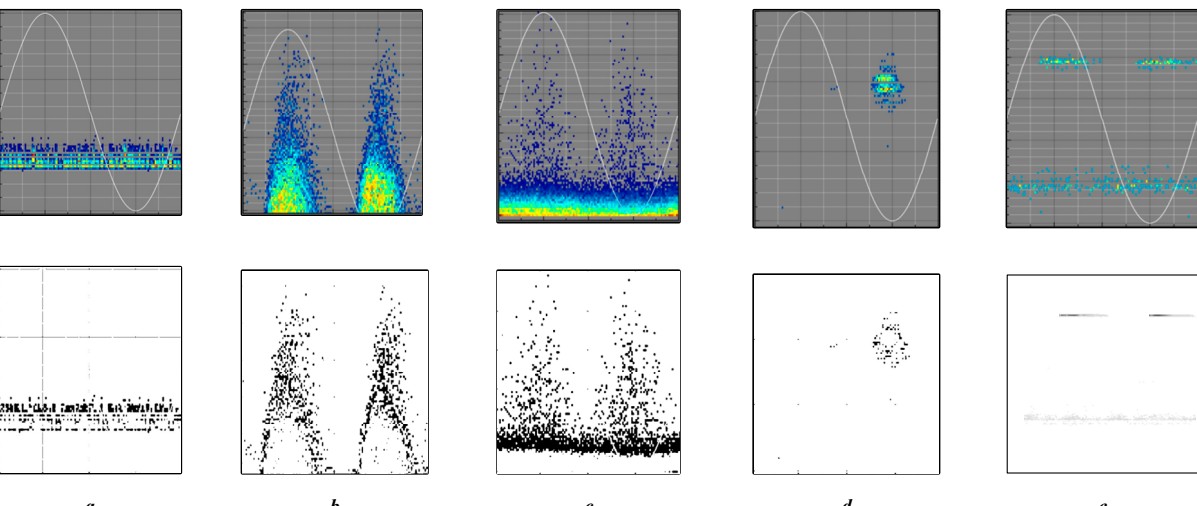

*a*           *b*           *c*           *d*           *e*

**Figure 3.** PRPD maps of grayness. (**a**) normal, (**b**) insulating void defects, (**c**) free metal particle defects, (**d**) metal tip defects, (**e**) floating electrode defects.

### 3. GIS Defect PD Pattern Recognition Based on YOLOv5 Algorithm

#### 3.1. YOLOv5 Detection Algorithm

YOLO series algorithm is a typical deep image recognition algorithm based on deep learning neural network, which has higher detection accuracy and needs less detection time compared with SSD and R-CNN target detection algorithms. Currently, the YOLO algorithm has been widely used in the field of transmission line image target detection in power systems. The YOLO algorithm was originally proposed by Redmon [12] and

others as a one-target detection algorithm. It transforms the target detection problem into a regression problem by dividing the picture into a grid and predicting the target by the grid. Subsequently, Redmon et al. [13] improved the YOLO algorithm and proposed the YOLOv2 algorithm, which uses Darknet-19 as a feature extraction network and improves the recall of the algorithm by introducing an anchor frame mechanism. In 2018, Redmon et al. [14] again improved the YOLOv2 algorithm by proposing the YOLOv3 algorithm, which changed the feature extraction network to Darknet-53 greatly improving the target detection efficiency and accuracy. YOLOv4 algorithm is a new target detection algorithm proposed by Bochkovskiy A et al. [15] in 2020. The backbone of the algorithm's CSPDarknet53 is obtained by combining cross stage partial network (CSPNet) with Darknet53, while adding pyramid pooling structure (SPP) combined with path aggregation network (PANet) to form YOLOv4 feature extraction network, which has a huge improvement in accuracy compared to YOLOv3.

YOLOv5 algorithm, as a representative of single-stage detection algorithm, has the advantages of simple procedures, small amount of code, fast detection speed, and high detection accuracy, has become the preferred option of target detection algorithm. The network structure of YOLOv5 is shown in Figure 4. YOLOv5 continues to use the CSP structure compared to YOLOv4, and adds the CSP structure to the backbone and Neck to enhance the feature fusion capability of the network. The backbone part of YOLOv5 uses the Focus network structure to perform slicing operations on the feature maps, allowing the input channels to be expanded by a factor of four, reducing the computational effort of the algorithm and increasing the computational speed. In the previous YOLO series, each real frame corresponds to one positive sample at training time, each real frame is predicted by only one prior frame at training time. However, in YOLOv5, in order to speed up the training efficiency of the model, the number of positive samples is increased, and each real frame can be predicted by multiple prior frames during training. Unlike previous versions of the YOLO series, YOLOv5 is available in four configurations: YOLOv5s, YOLOv5m, YOLOv5l, and YOLOv5x [16]. The main variations of these four structures are achieved by adjusting the depth multiplier and the width multiplier by setting different numbers of residual components in each cross stage partial networks (CSPN) to obtain networks of different depths, and setting different numbers of convolution kernels in the focusing structure and each CSPN to obtain networks of different widths. Table 1 shows the parameters of the YOLOv5 series network structure.

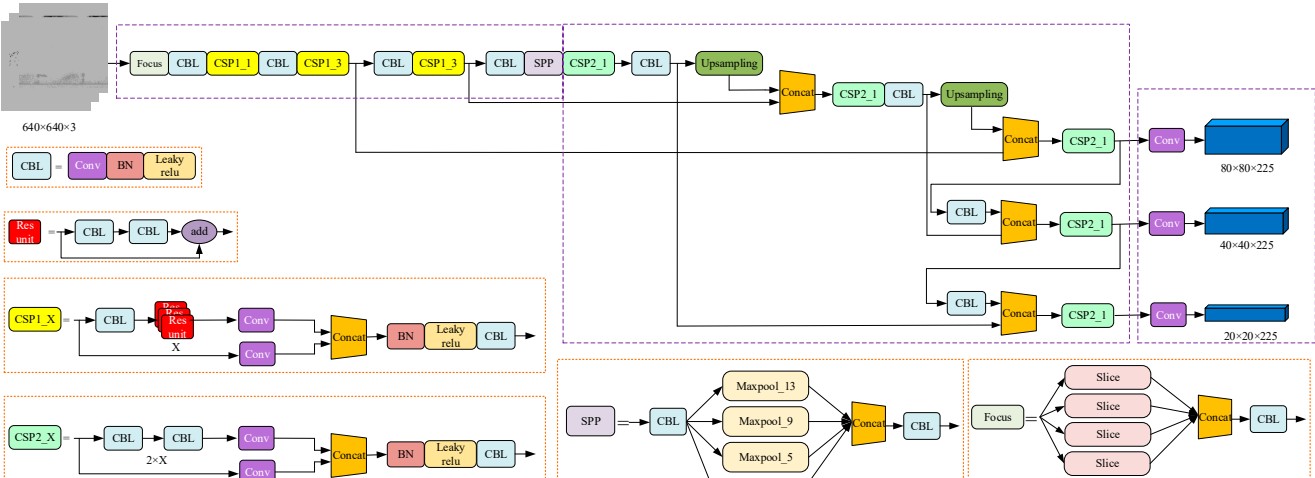

**Figure 4.** Flow chart of primary diagnosis based on deep forest.

**Table 1.** Parameters of the YOLOv5 series network structure.

| Parameters | YOLOv5s | YOLOv5m | YOLOv5l | YOLOv5x |
|---|---|---|---|---|
| Depth-multiple | 0.33 | 0.67 | 1.0 | 1.33 |
| Width-multiple | 0.50 | 0.75 | 1.0 | 1.25 |
| CSPN Number(Backbone) | 1, 3, 3 | 2, 6, 6 | 3, 9, 9 | 4, 12, 12 |
| CSPN Number(Neck) | 1 | 2 | 3 | 4 |
| Conv kernel number | 32, 64, 128, 256, 512 | 48, 96, 192, 384, 768 | 64, 128, 256, 512, 1024 | 80, 160, 320, 640, 1280 |

In this paper, YOLOv5m is selected as the target detection network for GIS partial discharge defects. YOLOv5m has a deeper network depth and larger network width than YOLOv5s, which increases the computational time but also increases the relative computational accuracy. Since GIS partial discharge defect map has no complex background to affect the detection effect, the use of YOLOv5m network can be well adapted to the construction of GIS partial discharge defect detection model.

### 3.1.1. Backbone Feature Extraction Network

The backbone feature extraction network used in YOLOv5 is the CSPDarknet network [17]. The use of a residual network is able to improve accuracy by adding considerable depth. Residual fast within YOLOv5 uses jump connections to mitigate the gradient disappearance problem associated with increasing depth in deep neural networks. A CSPN is introduced in the backbone feature extraction network, which splits the original stack of residual blocks into two parts, the left and the right. As shown in Figure 5, the trunk part continues the stacking of residual blocks, and the other part acts as residual edges, which are connected to the end between them after a small amount of processing.

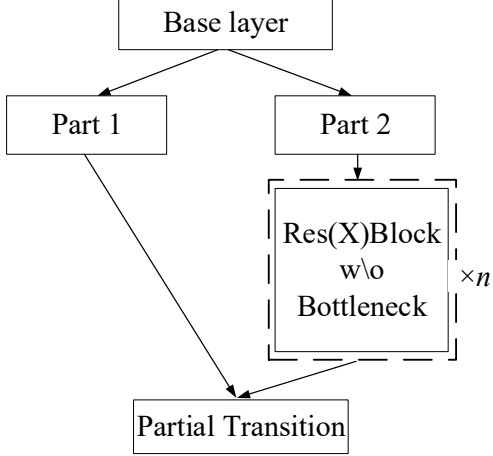

**Figure 5.** CSPNet network structure.

### 3.1.2. Focus Network Structure

In the process of YOLOv5 feature extraction network, the Focus network structure is used, and the Focus network structure is shown in Figure 6. The slicing operation is performed before the input image enters the backbone, as shown in the figure. Taking each image every other pixel, gives four images, four complementary images, and no information loss. In this way, the number of input channels is expanded by a factor of four, and the number of stitched together images is expanded to 12 compared to the original RGB three-channel pattern. The resulting image is convolved to obtain a two-fold down-sampled feature map with no information loss. The original $640 \times 640 \times 3$ image was transformed into a $320 \times 320 \times 12$ feature map by the Focus network image slicing operation. There are 48 convolution kernels in the YOLOv5m architecture, so one more convolution operation is performed to obtain a $320 \times 320 \times 48$ feature map. The slicing operation of the input image

by the Focus network structure can reduce the computation and increase the computation speed to some extent.

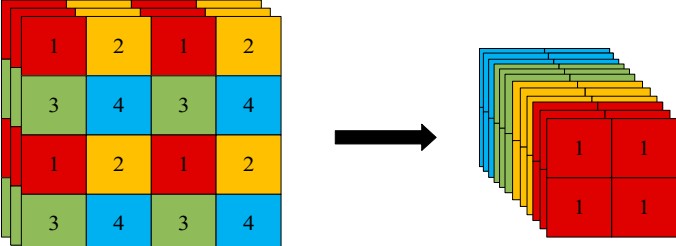

**Figure 6.** Focus Network Structure.

### 3.1.3. SPP Module and PANet Network Structure

The PANet network structure contains feature pyramid networks (FPN) structure, which achieves feature enhancement by fusion of shallow and deep features [18]. In YOLOv4, the Spatial Pyramid Pooling SPP module is used in the FPN. In YOLOv5, the SPP [19,20] module is used in the backbone feature extraction network. The structure of the SPP layer is shown in Figure 4. The maximum pooling layer Maxpool with different size pooling kernels ($5 \times 5$, $9 \times 9$ and $13 \times 13$) is taken for the maximum pooling of the extracted features. The bottom way is not pooled but directly connected. The results are channel stitched to increase the perceptual field of the network and extract the most important contextual features of the images.

In the feature utilization part, YOLOv5 uses multi-feature layer fusion for target detection, and a total of three feature layers need to be extracted. The three feature layers are located in the middle layer (CSP1_3), the lower middle layer (CSP1_3), and the bottom layer (CSP2_1) of the feature extraction network CSPDarknet network. When the input image of the feature network is $640 \times 640 \times 3$, the network shapes extracted by the three feature layers are $80 \times 80 \times 192$, $40 \times 40 \times 384$, and $20 \times 20 \times 768$ respectively. The construction of the FPN layer network is carried out after three effective feature layers are obtained. The feature pyramid can fuse the feature layers of different shapes for feature fusion and facilitate the extraction of better features [21]. The network structure of PANet is shown in Figure 7, P and N represent different feature layers. YOLOv5 applies the PANet structure in three effective feature layers: CSP1_3, CSP1_3, and CSP2_1, and achieves the fusion of feature information of the three scale feature layers.

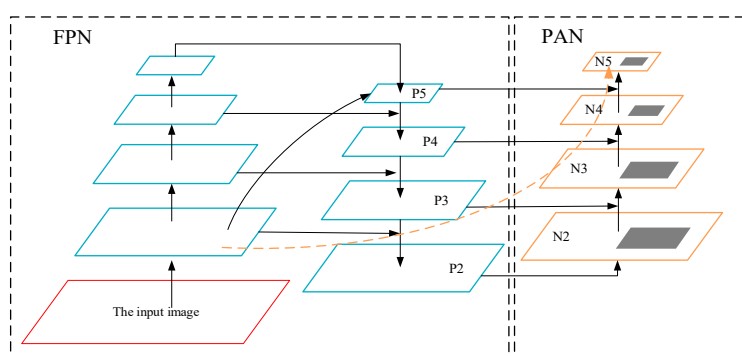

**Figure 7.** PANet network structure.

### 3.1.4. Activation Function and Loss Function

In the backbone feature extraction network of the YOLOv5 algorithm, the Leaky_relu activation function in the CBL convolution module in YOLOv4 is changed to the SiLu activation function [22]. The expression of the SiLu activation function is shown in Equation (1).

$$\begin{cases} f(x) = x \cdot \text{sigmoid}(x) \\ \text{sigmoid}(x) = x \cdot \frac{1}{1+e^{-x}} \end{cases} \tag{1}$$

where $x$ is the characteristic parameter. The SiLu activation function is a modified version of the Sigmoid and ReLu activation functions. The SiLu activation function is upper bound-free, smooth, and non-monotonic, and operates better than the ReLu activation function on deep models.

The selection of the loss function is crucial for evaluating the accuracy of target detection. The intersection over union loss (IoU Loss) [23] function used in the earlier YOLO series networks is simple in design, and only calculates the ratio of the intersection area and the union area of the prediction frame to the true frame, which often does not fully reflect the relative positions of the prediction frame and the true frame. Therefore, after introducing the two parameters of center distance and aspect ratio at the YOLO_Head position, the complete IoU loss (CIoU Loss) is formed as shown in the following equations:

$$\text{CIoU} = \text{IoU} - \frac{D_2^2}{D_C^2} - \alpha v \tag{2}$$

$$\alpha = \frac{v}{(1 - \text{IoU}) + v} \tag{3}$$

$$v = \frac{4}{\pi^2} (\arctan \frac{\omega_{gt}}{h_{gt}} - \arctan \frac{\omega}{h})^2 \tag{4}$$

where $D_2$ is the distance between the center point of the prediction frame and the target frame, $D_c$ is the diagonal length of the minimum external rectangle, $\omega_{gt}/h_{gt}$ and $\omega/h$ are the aspect ratios of the prediction frame and the target frame.

### 3.1.5. Prediction Network

GIS partial discharge PRPD map can be obtained with three enhanced features after passing through the backbone feature extraction network and the FPN feature pyramid. The shapes of these three enhanced features are (20, 20, 1024), (40, 40, 512), and (80, 80, 256). These three feature layers were passed into YOLO Head to obtain the prediction results. In YOLOv5, a total of nine a priori frames of different sizes are designed. Each feature point of each feature layer corresponds to three prior frames with dimension 9 (containing 4 prediction frame coordinates information, 1 confidence level, and 4 category information). The task of the prediction network is to select the target coordinates, confidence, and category information in the image.

The target is coarsely located using the a priori frame, and then the center point coordinates of the prediction frame $(b_x, b_y)$ and the width and height of the prediction frame $(b_w, b_h)$ are calculated by the following equation to determine the prediction frame position.

$$\begin{cases} b_x = \sigma(t_x) \times 2 - 0.5 + c_x \\ b_y = \sigma(t_y) \times 2 - 0.5 + c_y \\ b_w = P_w(\sigma(t_w) \times 2)^2 \\ b_h = P_h(\sigma(t_h) \times 2)^2 \end{cases} \tag{5}$$

where $\sigma$ is the sigmoid activation function, $t_x, t_y, t_w, t_h$ are the offset of the center point coordinates and the width-to-height scaling ratio of the network prediction frame, respectively, $c_x, c_y$ are the center point coordinates of the a priori frame, and $p_w, p_h$ are the width and height of the a priori frame.

When performing network inference, a scoring mechanism is applied to filter the most appropriate prediction frames. The class-specific confidence score of each target frame is obtained by multiplying the class information predicted by each grid with the confidence level of the target frame prediction, as shown in Equation (6).

$$Pr(C_i|O) \times Pr(O) \times \text{CIoU}_{\text{pred}}^{\text{truth}} = Pr(C_i) \times \text{CIoU}_{\text{pred}}^{\text{truth}} \tag{6}$$

where $Pr(C_i)$ is the category probability of each grid prediction, $Pr(O)$ is the target probability of each grid prediction, and $\text{CIoU}_{\text{pred}}^{\text{truth}}$ is the intersection ratio of the true frame to the predicted frame.

After getting the category confidence score for each target box, set the threshold value and filter out the target boxes with low scores. The retained target frames are subjected to non-maximal suppression (NMS) [24] to obtain the final detection results.

### 3.2. Model Training Method

In this paper, the GIS partial discharge map defective target detection model is trained using a migration learning [25] approach. To ensure the training efficiency of the model and avoid the problem of overfitting [26] due to insufficient training samples, the weight parameters are obtained by training on the migrated Pascal VOC public dataset. The model training process is divided into two stages: freeze training and unfreeze training. Freeze training can speed up the training speed and also prevent the weights from being destroyed at the early stage of training. Training starts from pre-training weights of the whole model with epoch set to 100 and freeze epoch set to 50. Freeze training is performed on the first 50 epochs of model training, and thaw training is performed on the last 50 epochs. The freeze training phase batch_size is set to 16, the thaw training phase batch_size is set to 8, and the maximum learning rate of the model is set to $1 \times 10^{-3}$. In addition, three training methods, namely Mosaic data augmentation, cosine annealing learning rate decay, and smooth labeling, are combined to improve and enhance the model training effect. YOLOv5 follows the Mosaic data enhancement approach of YOLOv4 [27]. The principle is to randomly read four images containing label annotations, and then stitch them to get one image by image broadening, which can effectively solve the situation of small target detection leakage in model training.

The learning rate decrease method used in this paper is the cosine annealing decay method, which is used to prevent the model from falling into local minima during training. The expression of the cosine annealing function is shown in Equation (7).

$$\eta_t = \eta_{\min}^i + \frac{1}{2}(\eta_{\max}^i - \eta_{\min}^i)[1 + \cos(\frac{T_{\text{cur}}}{T_i}\pi)] \tag{7}$$

where $\eta_t$ is the learning rate at moment $t$, $\eta_{\max}^i$ and $\eta_{\min}^i$ are the maximum and minimum values of the learning rate, respectively, $T_{\text{cur}}$ and $T_i$ are the current number of iterations and the total number of iterations of the ith training round.

For the multi-classification problem of GIS partial discharge defects, one-hot coding is usually used. When the predicted probability is used to fit the one-hot true probability, the generalization ability of the model may not be guaranteed to cause overfitting. Therefore, introducing label smoothing [28] training can better calibrate the network model, better generalize the network, and improve the model prediction accuracy.

$$q(K|x) = (1 - \varepsilon)q(x) + \frac{\varepsilon}{K} \tag{8}$$

where $q(x)$ is the one-hot label, $q(K | x)$ is the smoothed label, $K$ is the number of categories and $\varepsilon$ is the label smoothing value, which is generally a small constant and is set to 0.01 in this paper.

### 3.3. Model Performance Metrics

In terms of model recognition accuracy, the accuracy and recall rate are used as the basic indicators (the accuracy rate is mainly to assess whether the prediction is accurate, and the recall rate is mainly to assess whether the search is complete). The mean average precision (mAP) is calculated based on the accuracy and recall rates, and is used as the final evaluation index of accuracy to measure the combined performance of the trained models on all types. The mAP is the mean value of the average precision of the test results for all

categories, and the AP value is the area under the precision P and recall R curves. P and R can be calculated by Equation (9).

$$P = \frac{TP}{TP + FP}, R = \frac{TP}{TP + FN} \tag{9}$$

where TP is the number of correctly detected targets, TP + FP is the total number of targets detected by the model, and TP + FN is the total number of real targets. Taking the recall rate as the abscissa, and the maximum value of P under each recall rate as the ordinate, draw the precision-recall curve, and the area under the curve is the AP value. Suppose the total number of categories is *K*, and the mAP performance index can be calculated by Equation (10).

$$mAP = \frac{1}{K} \sum_{i=1}^{K} AP_i \tag{10}$$

The frames per second (FPS) are evaluated in terms of computing rate to examine whether the dynamic real-time monitoring requirements are met. The FPS value is greatly influenced by the performance of the equipment, and only this index is used in this paper for comparative analysis between models.

## 4. GIS Partial Discharge Defect Detection Algorithm

### 4.1. YOLOv5 Detection Process

The GIS partial discharge defect detection model uses the YOLOv5 algorithm to extract feature maps of three sizes: 20 × 20, 40 × 40, and 80 × 80, through a backbone feature extraction network. After the PANet up-sampling operation to achieve the fusion of deep and shallow information, the feature maps containing rich feature information in three different sizes are finally obtained. The basic process of GIS partial discharge defect detection using YOLOv5m in this paper is shown in Figure 8 and contains the following steps.

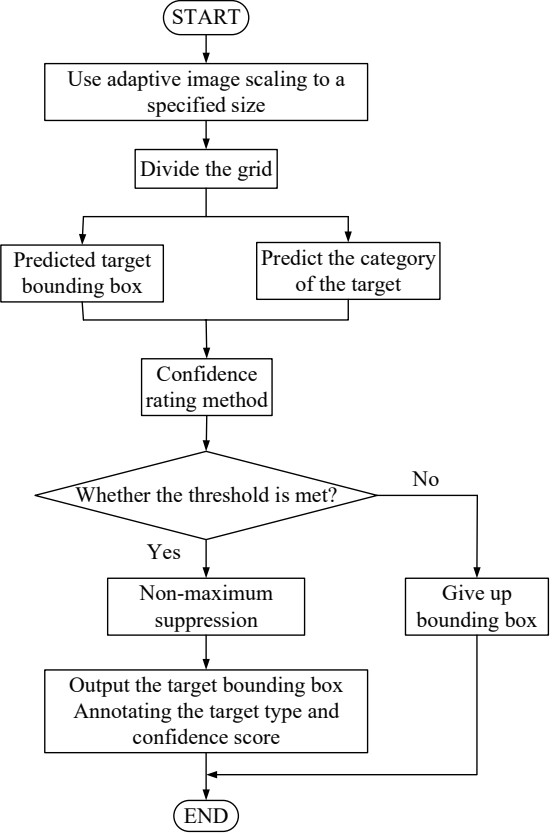

**Figure 8.** YOLOv5 target detection flow chart.

(1)  GIS partial discharge PRPD map is normalized to the prescribed input size of $640 \times 640$ using adaptive images and input to a target detection network for image feature extraction.

(2)  The target detection image has meshed, the target border and the classification to which the target belongs are predicted, and then whether the specified threshold value is met is judged according to the confidence score ranking.

(3)  The predicted edges that meet the specified threshold are retained, and the boundary edges generated by the detection are filtered using non-maximum suppression to eliminate redundant edges.

(4)  After eliminating all the redundant borders and marking out all the bounding boxes, output the target bounding box, label the target type and confidence score.

*4.2. Model Training and Testing Effects*

4.2.1. Test Environment Configuration and Data Preparation

The GIS partial discharge defect detection model is trained by migration using the weight parameters obtained from training on the Pascal VOC public dataset, including two stages of freezing and unfreezing. The parameters are set as shown in Table 2 below, and the environment setting parameters are shown in Table 3.

**Table 2.** Parameters for model training.

| Learning Step | Model Parameter | | | |
|---|---|---|---|---|
| | Training Sample | Batch-Size | Learning Rate | Epoch |
| Freeze | 3435 | 16 | $1 \times 10^{-3}$ | 50 |
| Unfreeze | 3435 | 8 | $1 \times 10^{-4}$ | 50 |

**Table 3.** Experimental environment configuration.

| Project | Environment |
|---|---|
| Operating System | Windows10 ($\times$64) |
| CPU | Intel Xeon E5-2678 v3 |
| GPU(MB) | NVIDIA GeForce GTX3080 Ti (16 G) |
| RAM | 64 G |
| Python | 3.9.1 |
| CUDA | 11.1 |

The image data used in this paper for target detection are all PRPD maps collected by simulating GIS partial discharge scenarios. The dataset format uses the Pascal VOC format and is manually annotated using the open source tool labeling. The labeled area of the sample is shown in Figure 9.

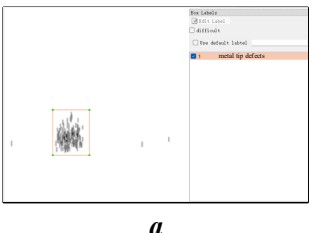
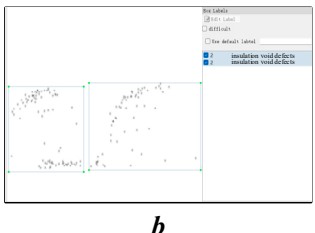
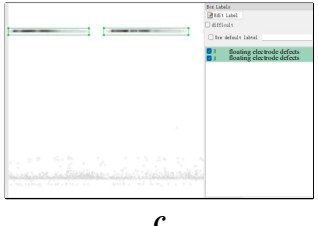
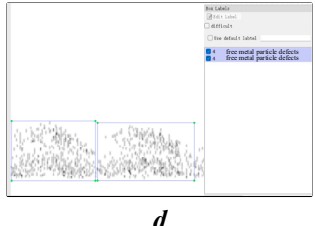

*a*  *b*  *c*  *d*

**Figure 9.** Sample markers of PD defects maps. (**a**) metal tip defects (**b**) insulating void defects, (**c**) floating electrode defects. (**d**) free metal particle defects.

4.2.2. Comparison of Training Methods

In this paper, different combinations of three training techniques of Mosaic data enhancement, cosine annealing decay, and smoothing labels, are used in the detection of GIS partial

discharge defects. According to the ratio of test set to validation set 9:1, 3435 samples were selected as the training set and 425 samples as the validation set to compare and analyze the model detection effect and detection efficiency under different training methods.

From Table 4, it can be seen that the best training results are obtained when both cosine annealing decay and label smoothing are used [29]. When the Mosaic data enhancement technique is used, a significant decrease in the mAP values of the training results can be seen. This is because Mosaic data enhancement increases the complexity of model training. The GIS partial discharge defect map background is relatively homogeneous and the defect features are relatively obvious, which makes the Mosaic data enhancement technique have a negative impact in GIS model prediction. Therefore, the optimal detection model can be obtained by using the technique of training with a combination of cosine annealing decay and label smoothing in the GIS partial discharge defect detection model training. The Loss decline curves for each Epoch during training and validation is shown in Figure 10. In 100 training rounds, the training loss values tend to decrease with the overall course of iterations. The loss values level off in the last few Epochs and the model gets the optimal weights. Using this model to detect the defects of GIS partial discharge, the mAP value can reach 95.87%.

**Table 4.** Comparison of training methods.

| Group | Training Methods | | | mAP/% | FPS |
|---|---|---|---|---|---|
| | Mosaic | Cos | Label-s | | |
| 1 | × | × | × | 94.28 | 25.36 |
| 2 | × | ✓ | × | 95.07 | 27.46 |
| 3 | × | ✓ | ✓ | 95.89 | 28.89 |
| 4 | ✓ | ✓ | ✓ | 91.73 | 26.53 |

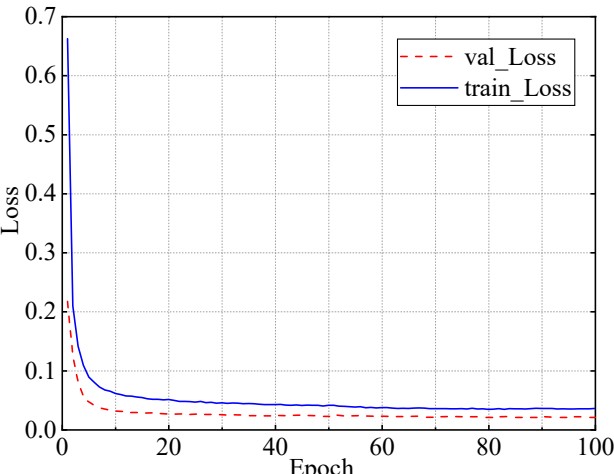

**Figure 10.** Loss curves for model training and validation.

By analyzing the effects of the detection environment and training techniques of the YOLOv5 algorithm on the testing effect of the GIS partial discharge defect detection model, the GIS partial discharge detection model under the optimal parameter settings was finally obtained. The partial discharge defects are detected using the test set obtained from the GIS partial discharge defect simulation experiments, and the detection results are shown in Figure 11. It can be seen that the YOLOv5 model proposed in this paper can effectively detect and identify the GIS partial discharge defect category. The time of partial discharge and the defect type of partial discharge can be clearly obtained through the detection results.

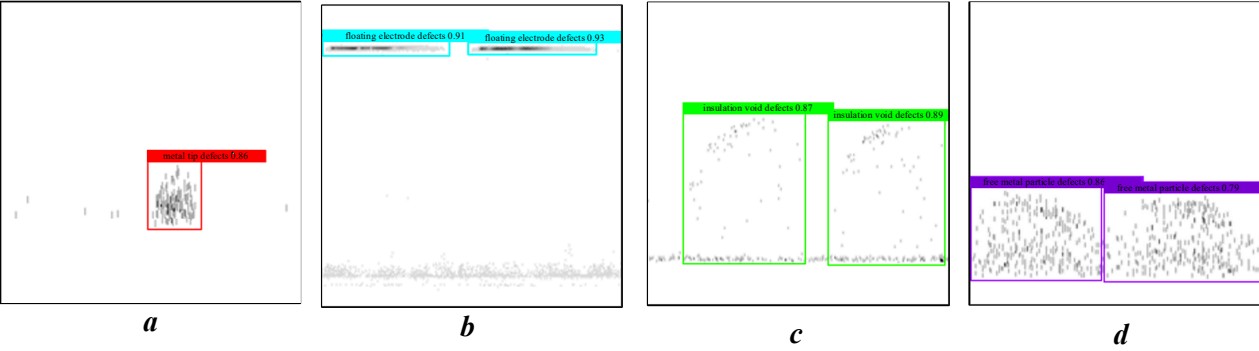

**Figure 11.** Result of PD defects detection. (**a**) metal tip defects, (**b**) floating electrode defects, (**c**) insulating void defects, (**d**) free metal particle defects.

*4.3. Comparison of Detection Methods*

GIS partial discharge defect detection models based on SSD, Faster-RCNN, YOLOv3, and YOLOv4 algorithms are established with the same training set and test set ratios to compare the metrics of different detection algorithms with the YOLOv5 model. The number of samples for the four types of partial discharge defects in the test set is 3435, and the number of samples in the test set is 425. The same validation set is used to verify the testing effect of different algorithms, and the detection efficiency and detection speed of each algorithm are judged by comparing the AP value, mAP value, and FPS of each type of algorithm. The detection results of different algorithms are shown in Table 5.

**Table 5.** Detection results of different algorithms.

| Method | Resolution | AP/% | | | | mAP/% | FPS |
|---|---|---|---|---|---|---|---|
| | | Corona Discharge | Insulation Discharge | Floating Electrode Discharge | Free Particle Discharge | | |
| SSD | $300 \times 300$ | 94.57 | 96.74 | 92.49 | 97.31 | 95.28 | 29.34 |
| Faster-RCNN | $600 \times 600$ | 97.20 | 98.71 | 82.81 | 95.73 | 93.16 | 18.42 |
| YOLOv3 | $416 \times 416$ | 93.57 | 96.06 | 95.99 | 96.17 | 95.45 | 27.39 |
| YOLOv4 | $416 \times 416$ | 92.88 | 70.51 | 90.17 | 68.21 | 80.44 | 27.45 |
| YOLOv5 | $640 \times 640$ | 96.74 | 97.01 | 93.76 | 96.06 | 95.89 | 28.89 |

By comparing the GIS partial discharge defect detection models under these five different algorithms, it can be seen that the YOLOv5 algorithm has the best detection results, both in terms of detection effectiveness and detection efficiency. Although the mAP value of SSD is comparable to YOLOv5 algorithm, and the FPS value is higher than YOLOv5 algorithm, the AP values of detecting the three defects of corona discharge, insulation discharge, and suspension discharge are far from acute for YOLOv5 algorithm. The AP value of YOLOv3 algorithm for corona discharge defect detection is not as good as YOLOv5 algorithm, and the FPS is also lower than YOLOv5. The YOLOv5 algorithm follows the optimized network structure of CSPNet and PANet for YOLOv4 network backbone feature extraction, and adds the Focus network structure to optimize the feature extraction algorithm, which not only improves the network computation speed but also improves the network computation accuracy. The YOLOv5 network is superior to the existing mainstream target detection algorithms in terms of both detection effect and detection speed.

The innovation of the GIS partial discharge defect detection model proposed in this paper is that the defect detection does not only tend to be classified, but can be monitored in real time during the practical application. In addition, when one or more types of faults occur at the same time, or when multiple faults occur within a short period of time, defect detection using the GIS partial discharge defect detection model proposed in this paper can be effectively identified and early warnings made in a timely manner to prevent serious

faults in GIS equipment and affect equipment operation. Multi-defect detection effect is shown in Figure 12. The situation when multiple types of defects appear one after another in a short period is represented by Figure 12a. The situation when suspension discharge and insulation discharge, metal particle discharge and insulation discharge, and corona discharge and insulation discharge appear simultaneously at the same time is represented by Figure 12b–d. From the detection results, it can be seen that the GIS partial discharge detection method based on YOLOV5 can effectively detect the results of multi-defective partial discharge, but the situation of leakage detection will occur. Therefore, in future research, we will strengthen the research on multi-defect partial discharge detection. The effect of multi-defect partial discharge detection will be improved.

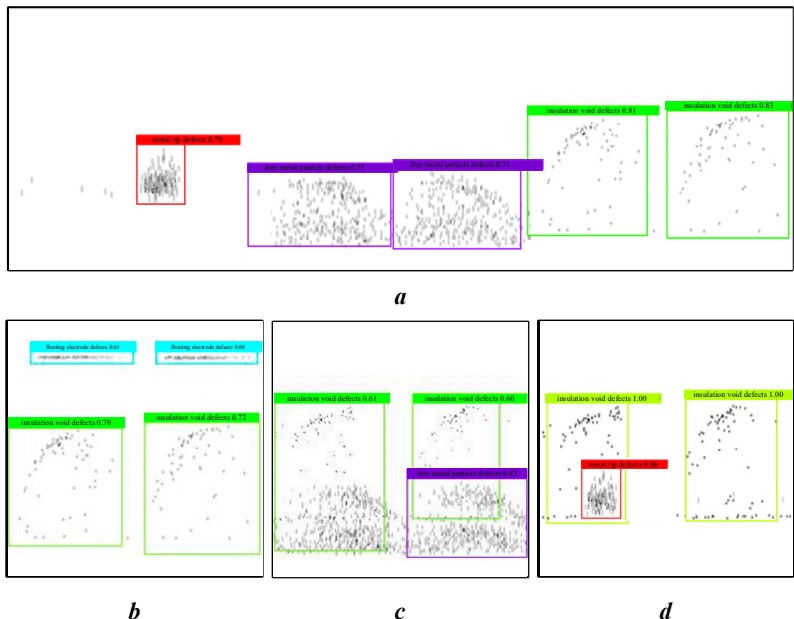

**Figure 12.** Multi-defect detection results. (**a**) metal tip defects, free metal particle defects and insulating void defects, (**b**) floating electrode defects and insulating void defects, (**c**) insulating void defects and free metal particle defects, (**d**) metal tip defects and insulating void defects.

## 5. Case Study

This chapter analyzes the effectiveness and practicality of the YOLOv5 algorithm in the field of GIS partial discharge defect detection. The trained GIS partial discharge defect detection model is applied to the field inspection.

The partial discharge online monitoring of a 330 kV GIS substation found that there was an abnormal UHF signal at the insulated basin location of a bus section. The signal is concentrated in the positive and negative half-period of the working frequency, and has certain symmetry, low number of discharges, low cycle repeatability, and scattered discharge amplitude. The partial discharge signal is monitored by adding UHF sensors to the insulated basin of the GIS busbar, and the PRPD map is obtained. The PRPD map signals of different locations and the identification results of GIS partial discharge defects are shown in Figure 13.

After determining the type of partial discharge and the location of partial discharge of GIS equipment, equipment disassembly and inspection were carried out. By inspecting each metal conductor and insulator piece by piece, the technicians finally found multiple obvious bubble-like projections on the concave side surface of the second basin insulator of busbar II, as shown in Figure 14.

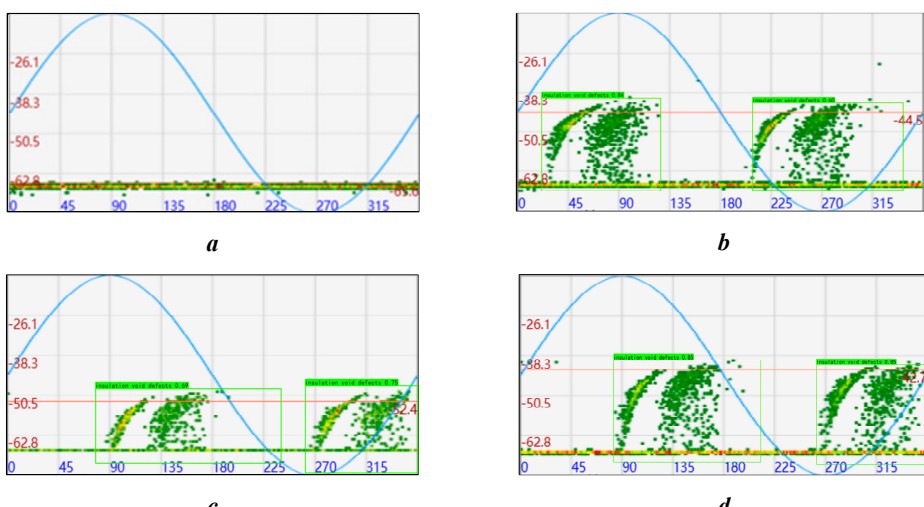

**Figure 13.** GIS partial discharge defect identification results. (**a**) channel 1, (**b**) channel 2, (**c**) channel 3, (**d**) channel 4.

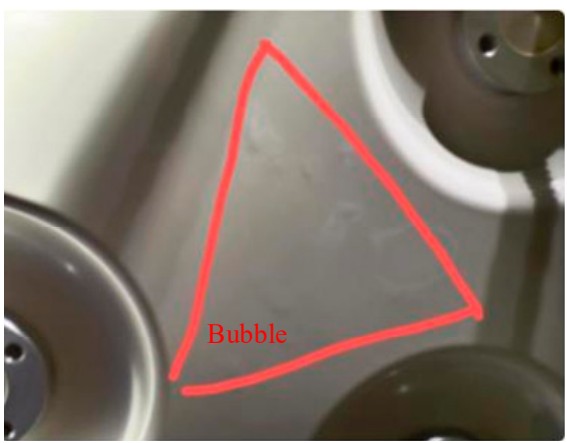

**Figure 14.** GIS disintegration pictures.

## 6. Conclusions

This paper proposes a GIS partial discharge defect identification model based on the YOLOv5 algorithm. After obtaining the partial discharge PRPD map dataset through GIS partial discharge simulation experiments, the map is grayed out to facilitate target detection and classification of the map. Different training methods are used to optimize the model, and the recognition model proposed in this paper is compared with other deep learning detection models. The recognition model is applied to practical application to verify the effectiveness and accuracy of the model. Some conclusions can be drawn as follows:

1.  The different combinations of three training techniques, Mosaic data enhancement, cosine annealing decay, and smoothing labeling, are performed. The detection result and detection efficiency of the models under different training methods are compared and analyzed. The optimal detection model can be obtained by using a combination of cosine annealing attenuation and label smoothing training techniques.
2.  The YOLOv5 model is compared with other deep learning target detection models. The results show that other models always have a lower accuracy rate in detecting a certain type of partial discharge defect, and the YOLOv5 model has higher target recognition accuracy and recognition result, with a mAP value of 95.89% and an FPS of 28.89.
3.  Based on laboratory verification results, the GIS partial discharge defect detection model can identify multiple fault types at the same time. It can be applied to the scene

where one or more fault types of GIS failure occur at the same time, and it can also be applied to the scene where GIS failure occurs in a short period of time.

4.  The GIS partial discharge defect identification model proposed is trained and used for practical application, and the consistency of the identification results and the disintegration results verifies the accuracy of the model.

Compared with traditional classification algorithms, the target detection algorithm can not only be used for real-time monitoring of GIS operation status, but also has high recognition accuracy for different types of PD defects. Multiple types of partial discharge defects can be detected, but due to the limited data of various types of partial discharge samples, the currently used multi-defective partial discharge detection samples are defective samples synthesized by image processing, which have certain limitations. The GIS partial discharge detection method based on YOLOv5 can effectively detect the results of multi-defective partial discharge, but there may also be a case of missed detection. Therefore, in future research, research on multi-defect partial discharge detection will be intensified, and more partial discharge maps will be collected under multiple defect types, and the sample size of multi-defect partial discharge defects will be increased, the model will be trained, and the detection effect of multi-defect partial discharge will be improved.

**Author Contributions:** Conceptualization, Y.L. and Z.Q.; methodology, Y.L. and Z.Q.; software, Z.W.; validation, Y.L. and Z.Q.; formal analysis, Z.Z.; investigation, Z.Z. and T.L.; resources, C.L.; data curation, Z.Z. and C.L.; writing—original draft preparation, Y.L. and Z.Q.; writing—review and editing, Y.L., Z.Q. and C.L.; supervision, C.L.; project administration, Y.L. All authors have read and agreed to the published version of the manuscript.

**Funding:** This work was supported by Science and Technology Project of State Grid Jiangsu Electric Power Company Limited, No. B310EF215PP7 Mobile GIS equipment partial discharge monitoring technology based on UHF signal intelligent analysis.

**Institutional Review Board Statement:** Not applicable.

**Informed Consent Statement:** Not applicable.

**Data Availability Statement:** All data used in this research can be provided upon request.

**Acknowledgments:** Yun Teng served as scientific advisors and critically reviewed the study proposal; Tianxin Zhuang participated in writing or technical editing of the manuscript.

**Conflicts of Interest:** The authors declare no conflict of interest.

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
