# Peer review of "A GIS Partial Discharge Defect Identification Method Based on YOLOv5"

_applsci, doi:10.3390/app12168360_

Round 1

Reviewer 1 Report

The result is comparable and good..

Please address the following

1. Explain one target detection algorithm, as YOLO was used to detect multiple object

2. Explain the migration learning, its reference is not accessible

3. Provide the link for migrated Pascal VOC public dataset

4. Explain thaw training and provide a reference for that

5. Precision and recall are sensitive to number of sample. Have you make them equal for each classes.

6. YOlO in line 169,171 should be YOLO

7. CAPN in line 196 should be CSPN

8. FPN expansion should be in line 219 in place of line 234

9. Please check sentences properly

Reviewer 2 Report

comments 

1- Add More details on the purpose of the PD analysis using YOLO in the abstract.

2- the most of figures are not clear. they need improvement.

3- line 33, add a reference 

4- What advantage YOLOv5 was used in this work? add some advantage in the introduction.

Reviewer 3 Report

In this paper the authors have utilized the YOLO algorithm on PRPD patterns to distinguish four different types of defects in GIS. This method also detects the multiple defects at the same time during practical application. However, there are some issues to address.

1. The abstract uses some abbreviations which are not clear to understand. Also, check the whole draft and give full forms before using abbreviations.

2. In page 3, Figure. 1(c) shows an insulating void defect. And the above paragraph said that it was made by using insulating tape to attach a fine metal wire of 1 mm in length to the insulator surface. Is this statement correct? 

1. The abstract uses some abbreviations which are not clear to understand. Also, check the whole draft and give full forms before using abbreviations.

2. In page 3, Figure. 1(c) shows an insulating void defect. And the above paragraph said that it was made by using insulating tape to attach a fine metal wire of 1 mm in length to the insulator surface. Is this statement correct?   I cannot understand how to produce a void in the insulation using tape and a fine wire.

3. In page 3, Figure 2. The spelling of “Measuring imbedance” is wrong.    In addition, it doesn’t work if the coupling capacitor located among two transformers. 

4. Authors should discuss more about the preparation of dataset from the defect models.

5. In abstract and introduction mentioned that the purpose of this paper is to distinguish multiple types of partial discharge defects. But the multi-defect detection effect was only mentioned in Figure 12. The relating statement is very less. And the PD patterns in Figure 12 is not reasonable because PD signals from multi-defect should not appear so separately in different voltage phase angle. Then, the comparison of different algorithms for multi-defect detection is also not included. So, it seems that this paper did not achieve the goal of the purpose. 

6. How to use the results from the laboratory PD measurements with conventional method and noise free environment? Please comment in the text.

7. In conclusion, you should critically discuss the advantages and limitations of your approach and how you like to overcome them in conclusion part.

Round 2

Reviewer 3 Report

Thanks for your quick response. I still concern about one question. There are two different statements about insulation discharge defect in revised draft. First one is wire inside insulator but second one is bubble inside the insulator. Their original statements are shown as follows. The first one is “… insulation discharge defect is simulated by placing thin fine wires of different sizes inside the insulator” in page 3; And the second one is “the insulation discharge defect type simulation, an insulator with simulated bubble diameters of 1 mm and 2 mm ...” in page 5. ? Which one is correct? This issue should be modified or explained before this paper be published. 
